# Impulsivity and Stillness: NADA, Pharmaceuticals, and Psychotherapy in Substance Use and Other DSM 5 Disorders

**Kenneth Carter and Michelle Olshan-Perlmutter\***

Department of Psychiatry, Carolinas Healthcare System, 501 Billingsley Road, Charlotte, NC, 28211, USA, E-Mail: kenneth.carter@carolinashealthcare.org

**\*** Author to whom correspondence should be addressed;
E-Mail: michell.olshan-perlmutter@carolinashealthcare.org; Tel.: +704-358-2722;
Fax: +704-358-2945.

**Abstract:** Pharmaceuticals and psychotherapy are commonly used in the management of impulsivity. The National Acupuncture Detoxification Association (NADA) protocol is an adjunctive therapy that involves the bilateral insertion of 1 to 5 predetermined ear needle points. One of the main benefits reported by patients, providers, and programs utilizing NADA is the sense of stillness, centering, and well-being. The induction of this attitude is seen as contributing to improved clinical outcomes including engagement and retention. The attitude of stillness is also suggestive of a pathway to mitigating impulsivity. Impulsivity is associated with substance use disorders and other DSM 5 diagnoses. Impulsivity has characteristics that are manifested clinically in behaviors such as disinhibition, poor self-control, lack of deliberation, thrill seeking, risk-taking. NADA holds promise as a useful treatment adjunct in the comprehensive management of disorders for which impulsivity is a prominent component.

**Keywords:** NADA; impulsivity; acudetox; mindfulness; pharmacology; adjunctive treatment; psychotherapy; substance use disorder; DSM

## 1. Introduction

DSM 5 is the standard classification used by mental health professionals and contains a listing of diagnostic criteria of every psychiatric disorder recognized by the US healthcare system. It is widely recognized that treatment modalities are not diagnosis specific and may be effective across diagnostic categories. Recent research has begun to focus on the putative relationship that exists across diagnostic categories, common behaviors, and shared neurobiologic substrates. This paper will explore the use of NADA (National Acupuncture Detoxification Association) protocol along with pharmaceutical and psychotherapy treatment modalities in the management of impulsivity. NADA is identified as a best practice in the acute and chronic management of substance use disorders since 2006 [1]. There is also growing evidence that NADA is useful across diagnostic categories where impulsivity is an important behavioral consideration.

## 2. Impulsivity and DSM 5 Diagnoses

Impulsivity has characteristics that are manifested clinically in behaviors such as disinhibition, poor self-control, lack of deliberation, thrill seeking, risk-taking [2]. Impulsivity is identified as a feature of many DSM 5 diagnoses. These include the substance use disorders, bipolar disorder, borderline personality disorder, antisocial personality disorder, intermittent explosive disorder, as well as gambling disorder, kleptomania, compulsive buying, and internet addiction [3,4]. Impulsivity comprises motor impulsiveness (acting without thinking), non-planning impulsiveness (a lack of forethought) and attentional impulsiveness (an inability to focus attention) [5,6]. Impulsivity is a predictor of treatment failure in substance use disorder [7–9]. High trait impulsivity in bipolar disorder is associated with more frequent episodes of illness, a history of suicide attempts, and a more severe course of illness [10]. Impulsivity is associated with similar problems in cluster B personality disorders including antisocial personality disorder and borderline personality disorder [11]. Suicidality in patients with bipolar disorder suggests a close association with impulsivity; in fact, impulsivity is highest in patients with the most medically severe suicide attempts [12].

Impulsivity can adversely affect the safety of social and occupational functioning, criminal activity and legal involvement, the safety of self and others. Modifying the risks associated with impulsivity can have significant impact on clinical outcomes.

## 3. Neurobiology and Impulsivity

In exploring the relationship between impulsive aggression and specific brain regions certain patterns emerge. Studies of impulsive aggressors have observed hypoactivation of the anterior cingulate cortex (ACC) and orbital prefrontal cortex (OFC) regions of the medial prefrontal cortex [13]. Structural MRIs in a subset of individuals with temporal epilepsy and impulsive aggression revealed either severe atrophy of the amygdala or lesions of the amygdala [14]. In Miller's literature review he reports on the hypothesis that the ACC and OFC in combination with abnormalities of the amygdala underlie the hyperarousal/dyscontrol states observed in impulsive aggressors [13]. Executive abilities (e.g., deductive reasoning, cognitive restraint of aggression, cognitive modulation of emotion and/or reflective functioning) are important in the regulation of aggressive impulses; executive abilities may themselves

be mediated by language processing regions. Diminished capacity to process and interpret complex language may thus contribute to impulsive aggression [13]. Impulsive aggression may be caused by multiple etiologies that continue to be investigated.

## 4. Treatment Option: Medication

Recent research suggests that impulsive aggression is modulated by emotional experience along with language, thinking, and behavior. Utilization of medication may provide benefit in addressing the aspects of impulsive aggression that are neurochemically mediated. Many medicines have been used in an effort to mitigate impulsivity associated with DSM 5 diagnoses. However, there are few controlled studies evaluating medication interventions in patients with impulsive aggression. Dilantin (phenytoin) and Prozac (fluoxetine) are thought to be associated with improvements in impulsive aggression and disturbances in the language processing regions of the brain [15–17]. Dilantin significantly reduced impulsive aggressive acts, but not premediated aggressive acts, among inmates in a Texas prison system; in this study, impulsive aggressors differ from nonimpulsive aggressors in their ability to decode verbal symbols (e.g., in reading) [15].

Mood stabilizers (anticonvulsants/Lithium) have been used off-label in the treatment of impulsive aggression; Lithium demonstrated a significant reduction in aggressive behavior in men who were prisoners in a medium security institution [18]. Trileptal (oxcarbazepine) appears to benefit adults with clinically significant impulsive aggression [19]. Lamictal (lamotrigine) appears to be a safe and effective medication in the treatment of anger in women with borderline personality disorder [20]. Topamax (topiramate) appears to be effective in the treatment of anger in women and men with borderline personality disorder [21,22]. Depakote (divalproex) shows evidence of effectiveness in treating impulsive aggression and irritability in patients with Cluster B disorders [23]. Risperdal (risperidone) alone or in combination with mood stabilizers may provide benefit in treating children and adolescents with aggressive behavior [24].

Intermittent explosive disorder is a disorder of impulsive aggression in which Prozac has been used; Prozac is thought to be effective because it enhances central serotonergic activity thereby reducing impulsive behavior [25]. Prozac is also associated with decrease in impulsive aggression in individuals with personality disorder [26].

Though the medications described above may have potential benefit, they also have the potential for significant short and long term side effects. Dilantin and the mood stabilizers side effects can include neurologic impairment including tremor and confusion, GI distress, and fetal abnormalities just to name a few. Prozac can cause GI distress, sexual dysfunction, sleep disturbance. Risperdal (a second generation antipsychotic agent) can dramatically impact metabolic function including weight gain, glucose dysregulation leading to diabetes and lipid abnormalities.

## 5. Treatment Option: Mindfulness Based Therapies

Mindfulness Based Therapies (MBT) are based on ancient healing practices. Mindfulness is a multidimensional practice that includes: (a) observing one's sensations, emotions, and thoughts; (b) describing one's emotions; (c) acting with awareness with focus on attention to the current task; (d) non-judging of inner experience without being critical of one's self, and; (e) non-reactivity to inner

experience allowing thoughts and emotions to come and go without reflecting too heavily on them or reacting to them [27].

There is a growing trend for the incorporation of Mindfulness Based Therapies (MBT) in the treatment of impulsivity associated with substance use disorder and other psychiatric disorders. There are significant associations between impulsivity and mindfulness. A study examining the relationship between impulsivity, mindfulness, and alcohol misuse concluded that the variable most significantly associated with alcohol misuse is negative urgency (*i.e.*, a proneness to act out under conditions of negative affect). With regard to drinking related consequences, lack of premediation and negative urgency is associated with adverse outcomes [28]. Individuals using MBT may learn to observe their thoughts and feelings without reacting to them; this facilitates better management of negative affect associated with impulsive substance misuse. One specific MBT technique that may help individuals manage strong impulses associated with addictive behavior is urge surfing. Urge surfing teaches clients to visualize an urge as a wave and to "surf the urge by allowing it to pass without being wiped out by giving in to it" [29]. MBT is also being used in addiction treatment to target craving, negative affect, and relapse prevention [30]. MBT may lead to greater attentional and inhibitory control by teaching clients to observe challenging emotional or craving states without habitually reacting [31]. These skills can be practiced regardless of the underlying cause of negative affect, *i.e.*, whether the cause of negative affect is a drug induced or a primary mood disorder [30].

## 6. Treatment Option: NADA Ear Acupuncture Protocol

In Sanscrit, the word "nada" has two distinct definitions: the primordial sound before words are formed and the first step when entering a temple. In Spanish "nada" means nothing. These definitions evoke what NADA trainers and practitioners refer to as "the Spirit of NADA". The acronym "NADA" (National Acupuncture Detoxification Association) thus implies that individuals obtain benefit from the NADA protocol by doing "nothing", *i.e.*, by being "still". This "stillness" is at the core of what is sought after in addressing the counterproductive aspects of impulsivity. This "stillness" is also parallel to what is experienced by patients who participate successfully in mindfulness based therapies. In differentiating NADA from MBT, it is clear that NADA is more "passive" in that participation is nonverbal and does not require a learning curve. NADA can be more effective than MBT in early abstinence and early stage recovery when individuals are likely to have a more limited capacity to be still, to focus, to concentrate, and to learn and apply new information [32]. In this sense NADA and MBT are quite complementary. NADA differs from MBT in that it is a somatic tool. MBT can take many sessions for an individual to learn and practice effectively. NADA, due to its immediacy of effect, is especially useful in acute situations, such as in individuals experiencing physiological or psychological distress from substance use disorder, trauma, and other behavioral health disorders. It is not uncommon for NADA practitioner to easily treat 10 to 15 individuals over the course of a 45 min treatment session.

NADA is a safe treatment option that has minimal side effects. Possible side effects are a piercing sensation when needles are inserted, post treatment redness or tenderness. Some patients become drowsy. Rarely, a vaso-vagal fainting response due to postural hypotension may also occur—this is easily resolved by simply removing the needles and providing supportive care while the patient recompensates. Other effects may include a sense of warmth, heaviness, or sleepiness [33]. The

American Acupuncture Council reported in 2012 that "in 30 yrs there is no record of a claim ever being filed against one of our insured auricular acupuncturists" [34].

NADA is simple and easily taught; it is commonly referred as acudetox, ear acupuncture detoxification, five-point auricular acupuncture, and five-point protocol; it involves bilateral needle insertion at auricular Sympathetic, Shen Men, Kidney, Liver, and Lung points (see Figure 1). The points have been shown to stimulate neurophysiologic, biochemical, endocrine, emotional, and cognitive effects. It well established that acupuncture treatment can also stimulate encephalogram, glomerular filtration rate (GFR), blood flow and breathing rate, and endogenous opiate peptides (e.g., beta endorphins and metenkephalins) changes [35]. It is associated with changes in other neurotransmitter levels including adrenocorticotrophic hormone (ACTH), cortisol, serotonin, norepinephrine, and dopamine [35].

The mesolimbic dopamine system includes the nucleus accumbens and prefrontal cortex. There is support for the concept that the GABA (B) receptor system may impact the mesolimbic system in drugs abuse including alcohol. Research suggests the possibility that acupuncture may play a role in suppressing the reinforcing effects of alcohol by activating the GABA receptor [36].

In Traditional Chinese Medicine the lack of calm inner tone is described as a condition of empty fire or "xu huo" because the heat of aggressiveness burns out of control when the calm inner tone is lost. The empty fire condition represents the illusion of power, an illusion that leads the addict to more desperate use of chemicals. NADA helps restore inner control in patients with empty fire [37].

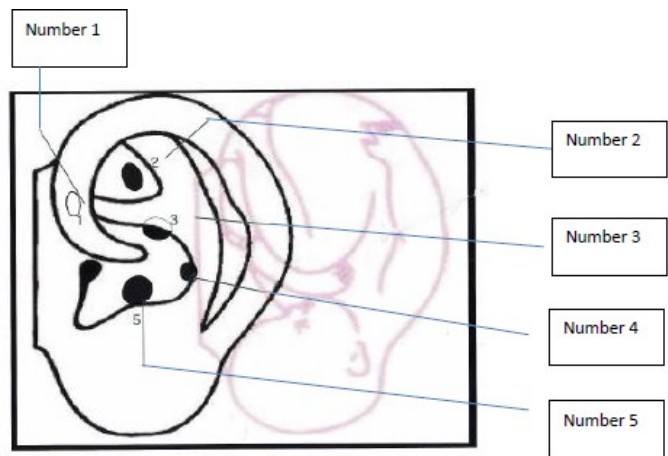

**Figure 1.** NADA ear and description. Number 1, Sympathetic; Number 2, Shen Men; Number 3, Kidney; Number 4, Liver; Number 5, Lung.

The arrangement of acupuncture points on the ear is similar to the arrangement of neurons in the motor cortex of the brain. The auricle is thus a "humunculus" representing the entire body in microcosm, it is a microsystem that reflects the body's state of health and wellness. Topographically the outer auricle can be represented as an inverted fetus [38] (see Figure 1).

Each of the 5 auricular points used in NADA is described uniquely. Sympathetic is related to disruption in both sympathetic and parasympathetic nervous systems; it has a strong analgesic and relaxant effect on internal organs as it dilates blood vessels. Shen Men regulates excitation and inhibition of the cerebral cortex and can produce sedation; it is used for many neuropsychiatric disorders. Lung is used for analgesia, sweating, and various respiratory conditions. Liver addresses symptoms associated

with poor liver functioning, neuralgia, muscle spasms and inflammation Kidney is the strengthening point for cerebellum, hematopoietic system; the kidneys and can relieve mental weariness, fatigue and headaches [39]. The NADA points can also be associated with various emotional states, *i.e.*, Shen Men with mental clarity; the Lung with the grieving process; the Liver with resolving aggression; the Kidney with willpower, coping with fear, and new growth [37].

NADA consists of the stimulation of 1 to 5 specified locations on the surface of the ear that alters and balances bodily function. The Chinese term for treatment location is "xue" which means opening. The traditional Chinese names for these locations often refer to flow on the surface of the earth such as valley, marsh, crevice, or stream. In the West the term "point" is used to identify treatment location. NADA points have less electrical resistance and, therefore, greater electrical conductivity. The points are warmer than the surrounding area by 0.1 to 0.2 of a degree. With NADA, as in acupuncture generally, precise point location varies within a small area. The location and function of these has remained consistent for centuries [39].

NADA originated at Lincoln Hospital in the Bronx, New York and gained prominence in treating addiction in the 1970s. In 2009 the NADA organization adopted a mission statement addressing a commitment to advocacy and training in "behavioral health including addictions". The 4th Edition of the NADA Resource Training Manual was adopted in 2010 [33].

White conducted a systemic literature review of the use of acupuncture in addiction treatment. Results were mixed regarding acupuncture benefit in abstinence, attrition, cravings, and withdrawal symptoms. There were notable disparities between patient reports of clinical benefit and the outcome measures selected by researchers in clinical trials. The disparities may be understood to be due, in part, to the failure to capture statistical separation between two active treatments, *i.e.*, the "active" and the so-called "sham" treatment. Studies using so-called "sham" acupuncture are less likely to be positive than those using non-acupuncture controls. Placing a needle any place on the body will elicit a response and thus "sham" acupuncture is not indeed a true sham that is an inactive control [40]. In fact, NADA is best understood, the authors believe, as a biopsychosocial intervention. In this holistic context, research is best conducted in the manner of naturalistic and qualitative studies—*i.e.*, by comparing [standard treatment alone] *versus* [standard treatment + NADA].

NADA is most efficiently provided in a group setting. A NADA practitioner can administer treatment for up to 20 individuals over the course of a 45 min treatment session. NADA as part of a comprehensive treatment strategy. There are over nearly 600 drug dependence rehabilitation facilities utilising NADA as part of their treatment program [40].

Given that NADA is most often provided in a structured group setting, patients joining the group are immediately introduced to a calm and supportive process that may help to mitigate impulsivity. Borderline personality disorder is characterized by impulsivity and emotional lability; patients BPD have consistently been associated with dropout from substance abuse treatment [41]. Stuyt's naturalistic study examined co-occurring substance abuse and borderline personality disorder (BPD). Those with BPD who received NADA were "able to sit still quietly without disruption and feel relaxed, which then enabled them to learn and practice more easily the other relaxation techniques being taught" [42]. Results indicated that the use of NADA was positively correlated with successful completion of the program (84%) compared with those not utilizing NADA (62%).

In another study, patients described their NADA response as a pleasant balancing experience, for example, "I was relaxed but alert" and "I was able to relax without losing control". Patients who were depressed or tired felt less depressed and more energetic, patients' general sense of well-being improved, and NADA facilitated a psychological readiness to benefit optimally from standard treatments offered in conjunction with NADA [32].

NADA is a nonverbal, nonthreatening intervention that often has an immediate calming effect on patients. This calming effect provides benefit for patients with an element of impulsivity across the spectrum of DSM 5 psychiatric disorders including the substance use disorders regardless of the specific substance [32,42]. Although there are no studies examining impulsivity as an isolated variable, factors associated with impulsivity improve with NADA. This includes negative affect, relapse prevention, reduced alcohol and drug cravings, and withdrawal symptoms associated with addictive substance use. NADA is consistently and reliably associated with improvement in engagement and retention [2,43]. In addition, the NADA is associated with a decrease in positive urine tests, increased program completion rate, improved patient satisfaction, and cost savings [44]. Multiple trials support the adjunctive use of NADA in treatment of heroin, cocaine and alcohol use disorder [45–49] and in anxiety states including those precipitated by trauma [47,50]. NADA has also been studied in a prospective trial showing improvement in symptoms that can be associated with impulsivity, namely anger, depression, impaired concentration, drug cravings, and physical discomfort [32].

## 7. Summary

Impulsivity is a central problem in the management of substance use disorders and is shared by many other DSM 5 disorders. Options available for managing impulsivity include medications, MBT, and NADA. NADA is a safe, efficient, inexpensive, and effective adjunctive treatment that compares favorably to other treatment options. NADA does not require verbal mediation and provides a quality of stillness that allows patients to be more "present" and thus psychologically available to benefit from the other biopsychosocial aspects of a comprehensive treatment program. Optimally, NADA is not used as stand-alone treatment, but instead as one element in a comprehensive treatment plan. The authors anticipate that NADA will be increasingly integrated into acute and chronic care settings where substance use disorders, co-occurring disorders, and other DSM 5 disorders are addressed.

Naturalistic and qualitative studies may be better at capturing the NADA benefits that patients report. Such approaches may bring additional insight into the discrepancies that exist between patients' positive reports and the mixed findings reported in the NADA literature.

## Author Contributions

Article written jointly by both authors.

## Conflicts of Interest

The authors declare no conflicts of interest.

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
