# Peer review of "Impulsivity and Stillness: NADA, Pharmaceuticals, and Psychotherapy in Substance Use and Other DSM 5 Disorders"

_behavsci, doi:10.3390/bs5040537_

Round 1

Reviewer 1 Report

This is a very nice, well written and easy to read overview of the problem of impulsivity and how it relates to multiple different diagnostic categories. It is a good review of the different treatment options available, pointing out how various medications can be helpful but also have the potential for significant side effects. It provides an excellent review of Mindful Based Therapies and why this is useful over a broad range of problems, teaching people ways to control their impulsivity. And it provides an excellent introduction to and overview of the NADA protocol and how it works in theory, including some treatment based evidence from patients' reports (as opposed to "evidence based treatment" from well-designed research studies). Since this is a difficult treatment to study in the "randomized double-blind placebo controlled" gold standard way, it is important to get this information out to the people working in the field so they can be motivated to incorporate it and then perhaps more research will result. As put by Adrian White "acupuncture may have some effects on drug dependence that have been missed because of choice of outcome in many previous studies, and future studies should use outcomes suggested by clinical experience". (White A. Acupunct Med 2013;0:18. doi:10.1136/acupmed-2012-010277)

The only problem I have with the paper is the inconsistent use of acronyms. I believe that the first use of the NADA acronym in the introduction should have (National Acupuncture Detoxification Association) after it, since this is the first place it appears in the body of the paper. After that it could remain NADA. However,  I am confused by the use of NP which is never spelled out? It seems to me that you mean this to be NADA and if so - all NP s should be changed to NADA. Otherwise, we would need an explanation for NP.

Also, I am curious as to why you have not included the reference: Stuyt EB. Acupunct Med 2014;32:318–324, since this directly relates to the use of the NADA protocol with people with borderline personality disorder, a population with a high degree of impulsivity.

Reviewer 2 Report

MAIN COMMENT

Articles about the NADA protocol are welcome in the literature.  However, this paper as submitted is puzzling and disappointing. 

The title is at variance with the content of the document - the title states the paper is about NADA in addictioncare.  The introduction states that "This paper will focus on NADA protocol as a therapuetic modality that is useful across diagnostic categories where implulsivity is an important behavioral consideration".  However, over one quarter of the paper focuses on medication and mindfulness based therapies for impulsivity, leaving less than a third of the paper to discuss NADA.

In its current form, the paper should more correctly be titled something like "Pharaceutical and non-pharmaceurtical treatment options for Impulsivity"

Furthermore, the current title focuses on addiction care, whilst the discussion about impulsive aggressiveness encompasses this condition in a much wider context than addiction alone.

If the authors' intententions are to focus on NADA's role in facilitating stillness as a means of addressing impulsivity in addiction care, then the paper needs to be restructured and rewritten to meet these intentions.

FURTHER COMMENTS AND SUGGESTIONS

Curiously, it appears that considerable effort has been made to reference all sections apart from the NADA section, which remains largely a description about what the NADA protocol is and how it might be applied.  While a cluster of studies are cited in lines 171-172, the rest of the NADA section consists of unsubstantiated generalisations and anecdotal information.  For example, almost every sentence in the paragraph starting at line 150 could and should be substantiated with references.

The NADA section needs to be expanded, rewritten to fulfil the title's promise, and substantiated with appropriate references. 

Furthermore, in the summary line 180, the claim is made that NADA is "safe, efficient, and cost-effective".  This is the first reference to safety in the paper.  If it matters to this paper, then it should be discussed fully and not casually introduced in the summary.

OTHER POINTS

Line 29: Should "compromises" be "comprises"?

Line 98: If this sentence remains in future versions, it requires editing as it currently begins "Individuals using MBT individuals may learn..."

Line 115: "patient's" should be "patients"

Lines 155-121: the authors discuss the relative merits of MBT and NADA, without mentioning that NADA requires someone to administer it.

Lines 131-138  Why is this information presented twice - once beneath the very poor diagram, and again in the paragraph commencing at line 143.  Also, there are more interpretations of the functions of the earpoints, such as Oleson's Auriculotherapy Manual, as well as others.

PLEASE avoid using a new acronym for NADA and do not refer to it as NP!  IT IS THE NADA PROTOCOL, OR NADA, AND NOT NP!

Round 2

Reviewer 1 Report

I do believe the manuscript has been significantly improved and believe it definitely warrants publication in Behavioral Sciences. The new title is more reflective of what the manuscript covers and the expanded information on NADA is helpful as this is the treatment option that will be the least known to readers. It is a well balanced and excellent overview of effective treatment options available for impulse control problems in a broad range of diagnositic categories. I believe this article will result in an increase in the use of NADA in acute and chronic care settings, as there is increased interest in adding tools that are effective but have less side effects than medications and NADA can be useful to help patients learn mindfulness based therapies.